# Hydrothermal Aging Mechanisms of All-Steel Radial Tire Composites

**DOI:** 10.3390/polym14153098

**Published:** 2022-07-30

**Authors:** Ning Jiang, Ru Zhang, Yuankun Li, Ning Li, Lingbo Dong, Chaozhong Chen, Cao Tan

**Affiliations:** 1School of Transportation and Vehicle Engineering, Shandong University of Technology, Zibo 255049, China; zhangru0816@163.com (R.Z.); m17685695474@163.com (Y.L.); m19862524906@163.com (C.T.); 2Research and Development Center, Triangle Tyre Co., Ltd., Weihai 264299, China; lining@triangle.com.cn (N.L.); donglingbo@triangle.com.cn (L.D.); 3Department of Technology Research, CRRC Academy Co., Ltd., Beijing 100036, China

**Keywords:** polymer matrix composites (PMCs), environment degradation, mechanical properties, cross-linking density, aging mechanism

## Abstract

This work focused on the effects of the hydrothermal environment on the aging of all-steel radial tire (ASRT) composites. Composite specimens were conditioned by immersion in deionized water at 30, 60 and 90 °C. Its water absorption, thermal and mechanical properties (tensile strength, elasticity modulus, elongation at break and interfacial shear strength), morphological structure, as well as molecular cross-linking reaction were investigated before and after aging. Results indicated that there was no dynamic equilibrium of water absorption of ASRT composites after deviating from the Fickian model. The molecular cross-linking density of the rubber matrix showed an increase in the early stage of aging. Then, the mechanical properties suffered of a drop due to the degradation of the rubber matrix and the poor interface between the steel fiber and rubber matrix. Additionally, a systematic hygrothermal aging mechanism was proposed.

## 1. Introduction

There is a growing interest in improving the durability and safety of tires. Due to advantageous properties [1,2,3] that include high elasticity, compression strength and good wear resistance, all-steel radial tire (ASRT) composites offer a huge alternative to conventional tire composites and are now widely used in heavy goods vehicles [4,5]. However, as the only part of the vehicle makes contact with the ground, the ASRT composites require sufficient mechanical properties to fulfill their function [6] and may be exposed to the hydrothermal environment. Therefore, it is important to understand how aging affects these composites, as it may limit their service life [7,8,9].

As the matrix of tire composites, rubber is generally considered a hydrophobic material, but temperature and moisture erosion will in fact affect its degradation [10,11]. Therefore, many studies focused only on the whole-tire durability studies in hot oxygen and ozone environments. Behnke et al. [12] conducted numerical research on thermal material degradation in steady-state rolling tires and conducted thermal mechanical coupling analysis of tires in steady-state motion through the finite element method. It was proved that the mechanical properties of rubber compounds change irreversibly with the increase in temperature, resulting in the change of tire structure. Luo et al. [13] established a complete simulation and test program to simulate the heat transfer characteristics of rubber during accelerated aging of tire. The simulation results show that the temperature difference between the surface and interior of a solid rubber wheel can reach four times. The results were compared with laboratory tests and showed good agreement. Wang et al. [14] conducted ozone aging on the new tire-tread rubber styrene-isoprene-butadiene rubber (SIBR). The changes of rubber microstructure at the ozone crack were monitored by micro-infrared technology, which confirmed that the rubber would produce cracks during ozone aging. Zheng et al. [15] studied the ozone-aging mechanism of natural rubber and found that the rubber molecular network was destroyed in the aging process, resulting in the weakening of mechanical properties.

Related to the works on the hydrothermal aging of tire composites, to our knowledge, few results have been put forward so far. Manaila et al. [16] investigated the water absorption of flax/natural rubber composites. The water diffusion was analyzed by Fick’s law, and the microcracks developed under the action of water molecules. Meanwhile, water absorption could bring internal stress due to the different expansion of fiber and rubber matrix, which in turn accelerated the hydrolysis of rubber composites [17]. Tires are complex structures with multilayer reinforcement. The stiffness and strength of tires largely depend on the performance of cord reinforcement [18,19]. Kenichi et al. [20] studied the degradation of rubber-to-brass adhesion by high-resolution photoelectron spectroscopy in a hydrothermal environment. On the contrary, Jeon et al. [21] stated that the copper plating on the steel wire can react with the rubber to produce sulfide during the vulcanization stage to promote interfacial bonding. To date, the effects of hydrothermal aging on the properties of ASRT composites, for example, the change of cross-link density and interface properties, are still not clear, which limited its applications.

In this study, the effect of hydrothermal aging on the water absorption, surface morphology, mechanical and thermal properties of ASRT composite were investigated. Molecular cross-linking density were tested to illustrate the aging degree of the rubber matrix. Finally, the hydrothermal aging mechanism of the ASRT composite was analyzed

## 2. Materials and Methods

### 2.1. Materials

In this study, rubber composite and ASRT composite, manufactured by Triangle Co., Ltd. (Shandong, China), and the manufacturing process are shown in Figure 1. The rubber composites were used to study the properties of rubber matrix in ASRT composites. Firstly, the natural rubber composites were mixed using a dense refiner. The dense refiner machine was preheated and the additives such as zinc oxide and carbon black were added in natural rubber. Then, the compound in mixing process was put into an open mill and the other additives (Table 1) were also added to refine rubber. The dumping temperatures of the masterbatch and refining rubber were 155 and 100 °C, respectively. The mixing times of the masterbatch and refining rubber were 180 and 120 s, respectively. Secondly, the mixture was calendered by S-type four-roller calender at a rate of 26 m/min under 95 °C to obtain thin sheets of rubber compound. Lastly, the calendered film was transferred to a plate vulcanizer and vulcanized at 150 °C to form a dense network structure to obtain rubber composites with natural rubber as the matrix and a variety of additives as the reinforcing phase.

Bundles of steel wire, consisting of 18 copper-plated steel wires with a diameter of 0.2 mm and a copper-to-zinc ratio in surface coating of 7:3, were added at the beginning of the calendering process to obtain ASRT composites, taking care to maintain the distance of steel wires, so that the composites can obtain the uniform internal structure. During the vulcanization stage, a good bonding interface between steel wire and rubber matrix were formed.

### 2.2. Hydrothermal Aging Experiment

Composite specimens were dried in a vacuum oven at 50 °C for 4 h prior to hydrothermal aging test. Then, the specimens were immersed in deionized water in test chambers (DK-8B, Shanghai Jinghong Test Equipment Co., Ltd., Jinghong, China) at 30, 60 and 90 °C, respectively. During the hydrothermal aging process, the specimens were removed periodically for different characterizations.

### 2.3. Water Absorption Test

The water absorption test was performed according to ASTM D5229. Test specimens (50 mm × 25 mm × 2 mm), cut from ASRT composites sheet, were weighed to test initial weight (*W*_0_) before aging using a high-precision balance (ZA120R4, Shanghai zanwei weighing machine Co., Ltd., Shanghai, China, precision = 1 mg). Weights (*W_t_*) of the specimens were recorded by periodic removal of the specimens from deionized water. The weight gain (M_t_) was calculated using Equation (1):(1)$$M_{t}=\frac{w_{t}-w_{0}}{w_{0}}\;\times \;100\%$$

All the values presented were averaged from five specimens.

### 2.4. Scanning Electron Microscopy (SEM) Analysis

The tensile fracture surface of NR composite specimens and water absorption surface of ASRT composite specimens were treated with golden powder, and then a scanning electron microscopy meter (Quanta-250, FEI, Hillsboro, OR, USA) was used to obtain micrographs at 30 kV accelerating voltage. All microscopy was performed at room temperature.

### 2.5. ^1^H-Nuclear Magnetic Resonance (NMR) Cross-Linking Density Analysis

The cross-linking density test of NR matrix of ASRT composite was carried out on XLDI-100 cross-linking density tester (Shanghai Newmark Electronic Technology Co., Ltd., New Taipei City, China), with resonance frequency of 15 MHz and magnet strength of 0.35 T. The test process was carried out at set intervals to take points for testing.

### 2.6. Fourier Transform Infrared Spectroscopy (FTIR) Analysis

Fourier transform infrared spectroscopy (FTIR) spectra for different specimens were recorded on a Nicolet5700 spectrometer (Thermo Nicolet Corporation, Madison, WI, USA) with the wavenumber ranging from 750 to 4000 cm^−1^.

### 2.7. Differential Scanning Calorimetry (DSC) Analysis

Differential scanning calorimetry (DSC) was performed on a dynamic Scanning Calorimeter (Q100, TA, New Castle County, DE, USA) at heating rate of 10 °C/min from −90 °C to 260 °C under nitrogen atmosphere with a frequency of 1 Hz.

### 2.8. Mechanical Properties Measurement

The mechanical properties of composites were measured using a Universal Materials Testing Machine (QJ211C, TASTE Instruments Co., Ltd., Suzhou, China), according to ASTM D3039 standard, with a cross-head speed of 500 mm/min. Dumbbell-shaped tensile specimens were cut from rubber composites. For pull-out test specimens, the copper-plated steel-wire fibers were embedded into rubber composites in a forming mold, with embedded steel-wire length of 25 mm (Figure 2). The specimens were then postcured at 150 °C for 55 min before they were removed from the mold. Steel-wire pull-out test was then conducted according to GB/T16586 at a cross-head speed of 100 mm/min and with a gauge length of 100 mm. The interfacial shear strength (IFSS) for each individual test was calculated by Equation (2).
(2)$$\tau =\frac{P}{\pi D_{f}L_{e}}$$
where *P* is the maximum pull-out load; the steel-wire fiber diameter *D_f_* and embedded length *L_e_* were measured with an optical microscope. About 20 specimens were tested for each group.

## 3. Results and Discussion

### 3.1. Water Absorption

Water absorption is an important parameter to evaluate the susceptibility of ASRT composites in hydrothermal aging. Figure 3a shows the relationship between weight gain (M_t_) of ASRT composites and time at 30, 60 and 90 °C. Figure 3b shows the M_t_ as a function of square root of immersion time divided by the thickness of the specimen. From the experimental data in Figure 3a, it can be seen that the M_t_ at different temperatures basically showed two-stage behaviors: a quick linear-increase initial part followed by a brief equilibrium plateau and a slow increase at last. Temperature showed a significant effect on M_t_; the values of diffusivity and equilibrium of water absorption increased with the increasing temperature. In order to determine the model of water absorption behavior, the curve fitting of Fickian law was illustrated in Figure 3b. It was found that the behavior of M_t_ of ASRT composites followed Fickian law at stage I (0~10 days at 30 °C, 0~6 days at 60 °C, 0~4 days at 90 °C), whereby water diffused into the composites without any physical change or chemical reaction [22]. This was supported by SEM observations (Figure 4a). The SEM observation of surfaces of water absorption samples shows no tendency for weakness between the steel wire and rubber matrix.

However, the experimental data deviated from the Fickian law after stage I; this was accompanied by the development of the poor interface between the fiber and matrix, which was indicated by SEM observation of interfacial failures (Figure 4b,d,f). This is because of the interstress caused by the different capacities of the steel wire and rubber matrix for water absorption and swelling in the ASRT composite [23]. The microcrack in the poor interface leads to further water absorption and a decline in interfacial bonding strength. As aging continues, the hydrolysis of rubber in ASRT composite brings a worse interface and the cracking of the rubber matrix (Figure 4c,e,g), leading to an increase in water absorption.

### 3.2. Cross-Linking Density Analysis

Cross-linking density is a crucial factor for molecular chain networks and has great influence on physical and mechanical properties [24]. Figure 5 shows the cross-linking density of the rubber matrix aging at three temperatures to analyze the reaction of internal molecules at different aging times. The cross-linking density of the rubber matrix at 30 °C presented a continuous increase. The value increased from 11.056 to 12.382 × 10^−4^ mol/mL after aging for 84 days, which suggested that the occurred cross-linking reaction of rubber increased the cross-linking and entanglement of molecular chains, resulting in the increase in the cross-linking density of materials [25]. This is because the temperature and water molecules enhanced the mobility of rubber chains as a driving force to facilitate cross-linking reaction. However, different from 30 °C, a decline in the cross-linking density of the rubber matrix was found after ageing for 84 days at 60 °C and 21 days at 90 °C. This is due to the hydrolysis of the rubber matrix, the rubber molecule chain break by the action of water molecules, oxygen and temperature. In addition, the different trends of cross-linking density value at three temperatures indicated that cross-linking reaction of rubber was strongly temperature-dependent. Temperature acted to speed up the aging of the rubber matrix, and therefore the cross-linking density at 30 °C would have a similar trend with that at 60 and 90 °C, based on the time–temperature equivalence principle [26].

In order to verify the reaction of rubber molecule chain, the main hydrolysis mechanism of rubber molecular chain is shown in Figure 6a, and FTIR results for the rubber matrix are presented in Figure 6b,c. In the spectrum (Figure 6b), the absorption peak at 831 cm^−1^, which was assigned to =C–H out of plane bending of the isoprene unit in the rubber molecules, was enhanced at 30 and 60 °C after aging for 49 days [27]. It can be interpreted as the increased numbers of =C–H due to the cross-linking reaction of rubber. The intensity of the absorption peak at 831 cm^−1^ decreased at 90 °C, indicating that the cross-linked rubber macromolecular chain was degraded after aging for 49 days, and the rubber molecular chain was broken and brought a weak peak. This is consistent with the results of cross-linking density analysis. Moreover, a new absorption peak at 1036 cm^−1^ for –C–O– stretching vibration appeared after aging for 49 days at 90 °C, which indicates that oxide was produced [28]. This also confirmed that the hydrolysis degradation occurred in the rubber matrix under the action of water and oxygen, as shown in Figure 6a.

Figure 6c shows the FTIR spectra of the rubber matrix at 90 °C for different aging times. From the spectrum, the bands at the peak of 1650 cm^−1^ are attributed to C=C stretching vibration in the rubber backbone structure [29]. The resonance peak intensity increased first and then decreased with aging time, which can be illustrated by the results of cross-linking density. The cross-linking degree of the rubber molecular chain increased, bringing a strong peak, then the bond of C=C was destroyed due to hydrolysis degradation, which showed a weak peak. This is consistent with the reported results of Wu [30], who studied the quantitative analysis of rubber composite aging using FTIR. In addition, the broad peak reflects that the OH stretching vibration in the region between 3300 and 3400 cm^−1^ is significantly enhanced with the increase in aging time. On one hand, the increased water absorption brings a strong OH bond vibration of water, which can be proved by the water absorption test. On the other hand, the deep hydrolysis degradation of the rubber molecule occurred during aging, which produced new hydrogen-bonded OH groups (Figure 6a).

### 3.3. Differential Scanning Calorimetry Analysis (DSC)

To further study the effect of hydrothermal aging on the thermal properties of ASRT composites, the glass transition temperature (T_g_) of the rubber matrix was investigated by differential scanning calorimetry (DSC) as shown in Figure 7. The inflection point with the largest slope in the glass transition region is defined as the T_g_ which is the dividing point between the glassy state or rubbery state of material. A tangent line was made at T_g_ to intersect the horizontal baseline before and after the change of the transition region. The intersection point was regarded as the start temperature and end temperature of glass transition, respectively, and the width of the temperature interval was defined as the width of the glass transition region, which was affected by the number of types of blends. The more kinds of blends in the rubber matrix, the wider the glass transition region will be.

A summary of DSC results is presented in Table 2 and Figure 8. It could be found that the T_g_ of the rubber matrix increased over time, while the width of the glass transition region reduced at 30 °C. This is clearly due to a hindered mobility of the molecules, because of the cross-linking reaction process. However, the T_g_ showed a decrease after 49 days of aging at 60 °C and 21 days of aging at 90 °C. This can be due to both the mobility enhancement caused by the plasticizing effect of more absorbed water and the smaller molecules resulting from the hydrolysis degradation process. This agrees with cross-linking density analysis results. In the corresponding width of the glass transition region, an evident increase was found after aging for 49 days at 60 and 90 °C. This is because the rubber molecular structure was destroyed by hydrolysis, and the transformation of the group structure leads to a richer variety of chain segments, which widened the glass transition region.

### 3.4. Mechanical Properties

#### 3.4.1. Tensile Properties of Rubber Matrix

Figure 9 shows tensile properties of rubber composites immersed in deionized water for different aging times at 30, 60 and 90 °C, respectively. There was a small decrease in the tensile strength of the rubber matrix after 14 days’ aging at 30 °C. The elongation at break increased. This was due to the plasticizing effect of water molecules; water acted as an effective plasticizer, which increased the aggregation of rubber chains [31,32,33]. There was no significant damage in rubber composites, which can be proved by SEM observations (Figure 10a,b). Then a decline in tensile strength and elongation at break was observed with aging time at 30 °C, which was due to poor interface between rubber and additives. A similar result has been reported by the study of Saijun [34]. The different capacities for water absorption and the swelling of rubber and additives led to internal stresses in the rubber matrix, which resulted in microcracking in the interface. This is indicated by SEM observation (Figure 10c). With the poor interface, stresses could not be transferred efficiently, leading to deterioration of the mechanical strength of the rubber matrix. As aging time increased, more microcracks appeared (Figure 10d) due to the development of a weak interface between rubber and additives, which led to further decline in the composite’s tensile strength and elongation at break.

The tensile strength and elongation at break of the rubber matrix dropped faster at 60 and 90 °C. This is because the propagation of microcracks at the interface was earlier than that at 60 °C, and especially at 90 °C. The high temperature assisted the active molecular motion, which in turn facilitated the diffusivity of water into the rubber matrix and promoted the development of a weak interface. This is proved by water absorption test and SEM observations (Figure 10f,i). There were further decreases in mechanical properties after aging for 56 days at 60 °C and 21 days at 90 °C. This is because the fragmentation of the macromolecule chain and the microstructural damage of the rubber matrix due to the degradation of rubber by the hydrolysis, as was consistent with previous cross-linking density analysis and SEM test (Figure 10g,j).

The Figure 9c showed the evolution of elasticity modulus of rubber composites with the increase in aging time. It can be seen that the modulus increased for the specimens being aged at 30 °C, while it slightly decreased for those aged at 60 °C and seriously decreased for those at 90 °C. The reason for the increase in modulus is the increased cross-linking density. Then, the structural damage and chemical degradation at 60 and 90 °C brought a decrease in modulus.

#### 3.4.2. Interfacial Properties of ASRT Composites

Figure 11 shows the change in the interfacial shear strength of ASRT composites before and after hydrothermal aging at 30, 60 and 90 °C. The shear strength had a slight rise at the beginning, then descended gradually at 30 and 60 °C. This is consistent with the results of Li et al. [35], who studied the effect of thermal and moisture aging on the adhesive properties of rubber and steel cord. Figure 12a shows a SEM image of the interface between the copper wire and rubber matrix of the ASRT specimen aged at different times under 60 °C. Before aging, a good adherence between the steel wire and rubber matrix was observed. Then, the CuS layer was formed in the interface due to the chemical reaction between copper and sulfur, which was on the steel wire surface and rubber matrix, respectively. Similarly, the ZnS layer was also formed in the interface by reacting in the same way at this stage. This is supported by energy spectrum analysis; large amounts of copper, zinc and sulfur elements were found in the interface (Figure 12d). With the connection of CuS and ZnS, the interface between the steel wire and rubber matrix was further enhanced after aging for 21 days at 60 °C, as shown in Figure 12b. Similar conclusions have been reported by previous studies [36,37]. As more water was absorbed by the specimens, the swelling of rubber increased, resulting in a large swelling stress which induced the damage at the interface (Figure 12c), leading to a decline in the interfacial shear strength of specimens being aged at 30 and 60 °C. Meanwhile, for the specimen aging at 90 °C, there was no increase in interfacial shear strength in the early stages of aging due to the higher water absorption and premature damage at the interface.

### 3.5. Hydrothermal Aging Mechanisms of ASRT Composite

According to analysis of water absorption, morphological, cross-link density and thermal and mechanical properties before and after aging at different temperature, the hydrothermal aging mechanisms of ASRT composites can be summarized in Figure 13. Firstly, water molecules diffused into the composites through the rubber matrix and the interface between the steel wire and rubber matrix; the behavior of water absorption of ASRT composites followed Fickian behavior. The water molecule acted as an effective plasticizer, which increased the aggregation of PLA chains and brought a small decrease in tensile strength. There was no observable physical change or chemical reaction at this stage (Figure 13b).

Then, the CuS and ZnS layer formed under the action of water and temperature at the interface between the steel wire and rubber (Figure 13c), thus promoting the effect on tire adhesion. The interface and interfacial shear strength of ASRT composites were improved.

Gradually, as aging continued, the cross-linking density of rubber increased due to the occurrence of cross-linking reaction, as shown in Figure 13d. The cross-linked network macromolecule brought the increase in tensile elastic modulus and the decrease in elongation at break. At the same time, microcracks formed at the interface due to the rubber swelling and the creation of internal stress, ultimately leading to a decline in the composite’s tensile strength and interfacial shear strength during this stage.

Finally, the cross-linking structure of rubber was destroyed by the hydrolysis process. It caused cracking of the rubber matrix (Figure 13e), which led to the increase in weight gain of the ASRT composites. In return, more water absorption promoted the hydrolysis of rubber and further increased the cracking of the rubber matrix and interface, which brought a further decrease in mechanical properties and eventually led to the failure of ASRT composites.

## 4. Conclusions

The effects of hydrothermal aging on the properties of ASRT composite was revealed by the investigation of water absorption behavior, microstructure, cross-linking density and thermal and mechanical properties. Four degradation stages of hydrothermal aging of ASRT composites are observed and can be related to the changes in mechanical properties. In stage I, there is a reduction in tensile strength due to the plastication of water molecules, and no observable change occurred in the microstructure. In stage II, the tensile strength and interfacial shear strength increase slightly due to the better interface caused by the generation of the CuS and ZnS layer. In stage III, both the cross-linking reaction and interface degradation take place due to the action of water and temperature, which leads to a decline in the composite’s tensile strength and interfacial shear strength but an increase in tensile modulus. In stage IV, hydrolysis of the rubber matrix causes microcracking in both the matrix and interface with a further significant decrease in mechanical properties.

## Figures and Tables

**Figure 1 polymers-14-03098-f001:**
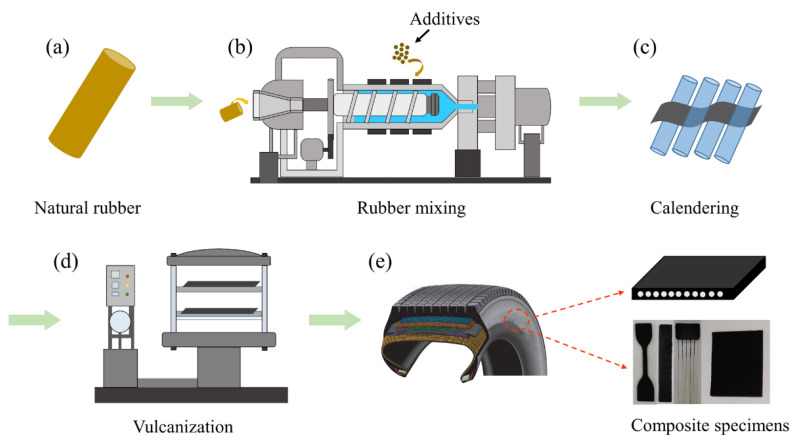
Manufacture route of rubber composite and ASRT composite specimens. (**a**) Natural rubber; (**b**) Rubber mixing; (**c**) Calendering; (**d**) Vulcanization; (**e**) Composite specimens.

**Figure 2 polymers-14-03098-f002:**
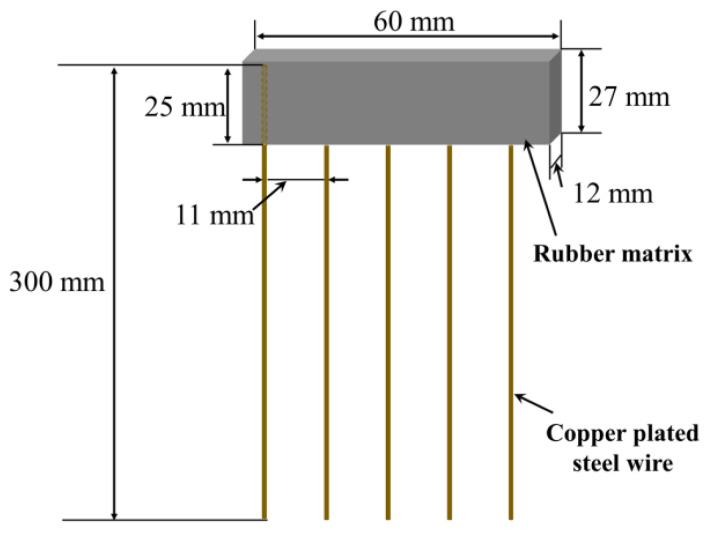
Steel-wire pull-out test specimen of ASRT composite.

**Figure 3 polymers-14-03098-f003:**
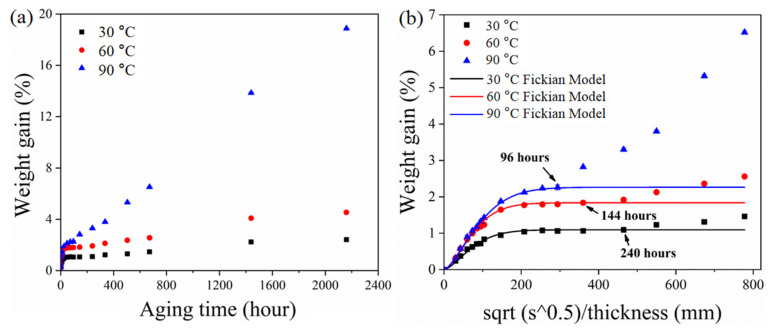
Water absorption curves of ASRT composites at different temperatures. M_t_ as a function of (**a**) time and (**b**) square root of immersion time divided by the thickness.

**Figure 4 polymers-14-03098-f004:**
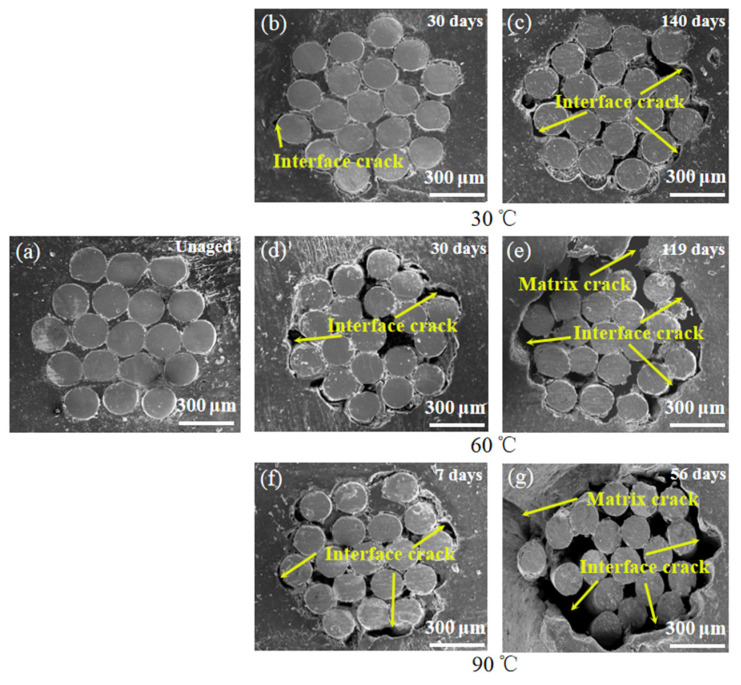
SEM image of steel/rubber composite interfaces after different aging times at 30, 60 and 90 °C. (**a**) 60 °C unaged; (**b**) aging 30 days at 30 °C; (**c**) aging 140 days at 30 °C; (**d**) aging 30 days at 60 °C; (**e**) aging 119 days at 60 °C; (**f**) aging 7 days at 90 °C; (**g**) aging 56 days at 90 °C.

**Figure 5 polymers-14-03098-f005:**
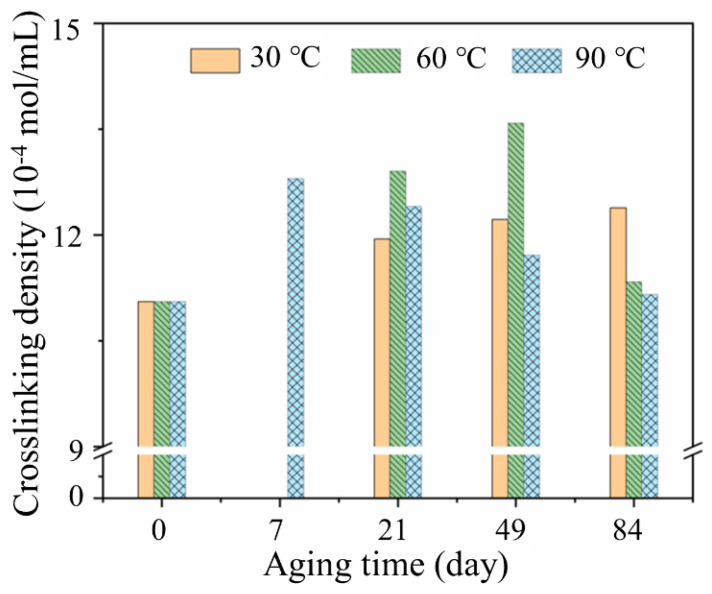
Cross-linking density of rubber matrix during aging at 30, 60 and 90 °C.

**Figure 6 polymers-14-03098-f006:**
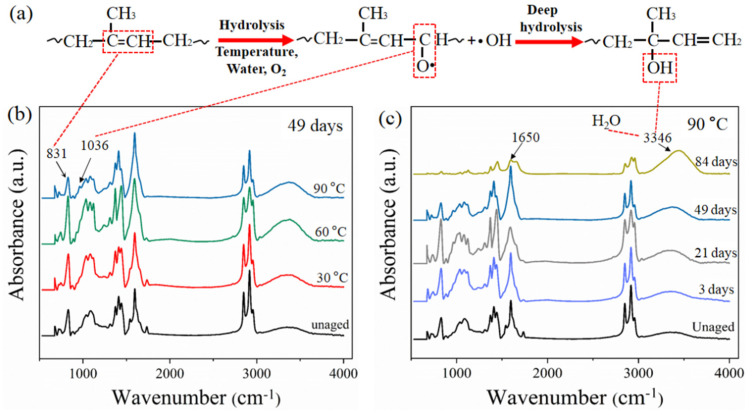
(**a**) Main hydrolysis mechanism of rubber molecular chain. FTIR spectra of rubber composites under different conditions: (**b**) Aging at different temperatures for 49 days; (**c**) aging at 90 °C for different days.

**Figure 7 polymers-14-03098-f007:**
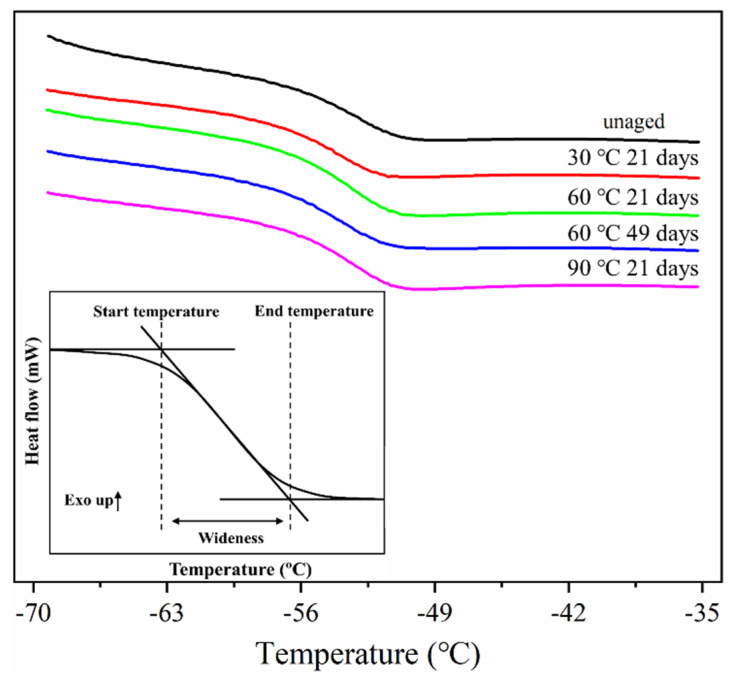
Schematic diagram of glass transition width of NR composite under different conditions.

**Figure 8 polymers-14-03098-f008:**
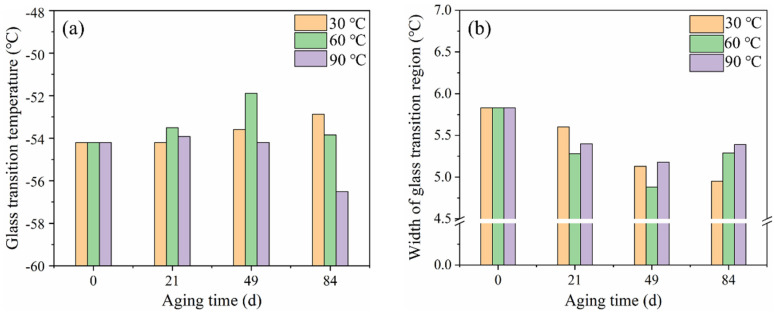
Evolution of (**a**) T_g_ and (**b**) width of glass transition region of NR composite with hydrothermal aging at different temperatures.

**Figure 9 polymers-14-03098-f009:**
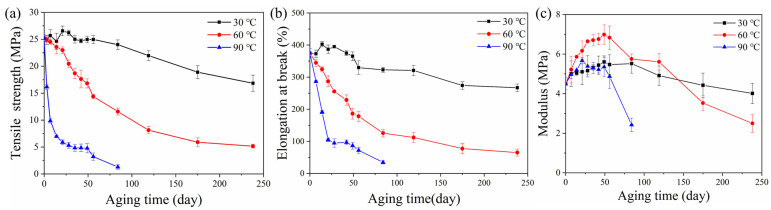
Mechanical properties of rubber composites hydrothermally aged at different temperatures: (**a**) Tensile strength; (**b**) elongation at break; (**c**) elasticity modulus.

**Figure 10 polymers-14-03098-f010:**
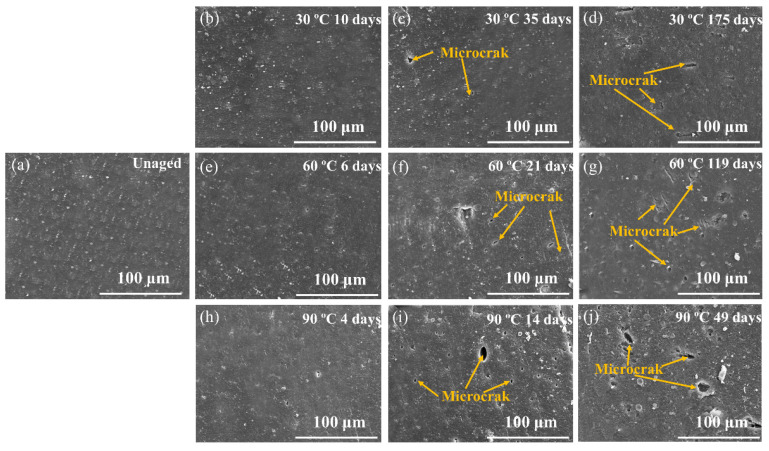
SEM image of the fracture surfaces of rubber composites at different aging times under 30, 60 and 90 °C. (**a**) 60 °C unaged; (**b**) aging 10 days at 30 °C; (**c**) aging 35 days at 30 °C; (**d**) aging 175 days at 30 °C; (**e**) aging 6 days at 60 °C; (**f**) aging 21 days at 60 °C; (**g**) aging 119 days at 60 °C; (**h**) aging 4 days at 90 °C; (**i**) aging 14 days at 90 °C; (**j**) aging 49 days at 90 °C.

**Figure 11 polymers-14-03098-f011:**
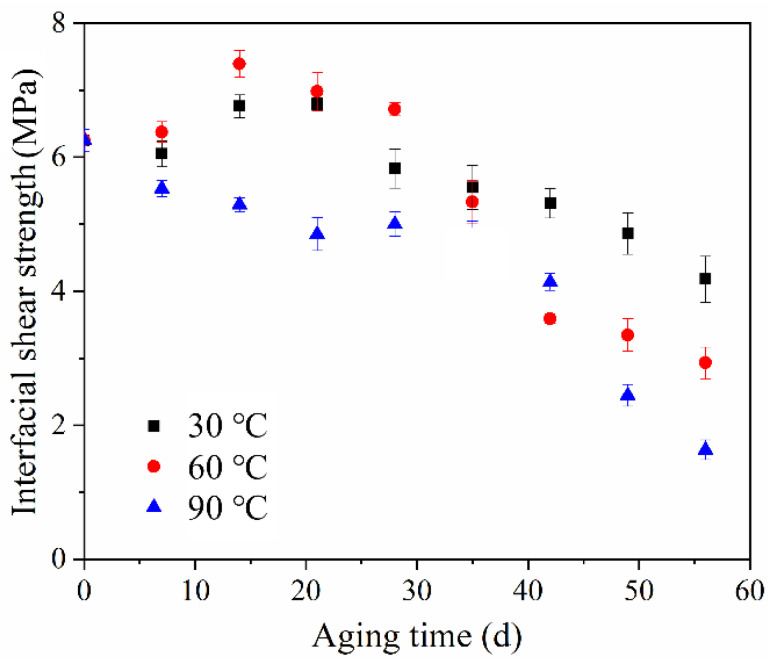
Interfacial shear strength of ASRT composites at different aging times under 30, 60 and 90 °C.

**Figure 12 polymers-14-03098-f012:**
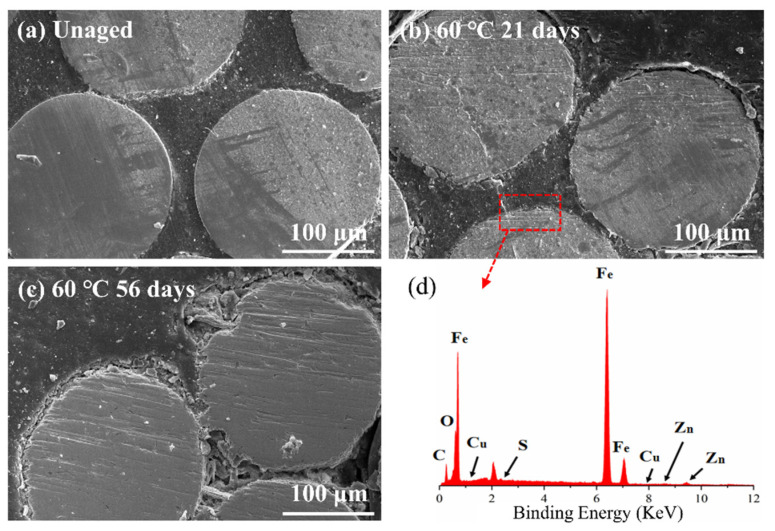
SEM image of the water absorption surfaces of ASRT composite (**a**) before aging; (**b**) aging 21 days at 60 °C; (**c**) aging 56 days at 60 °C; (**d**) Energy spectrum analysis of steel-wire surface.

**Figure 13 polymers-14-03098-f013:**
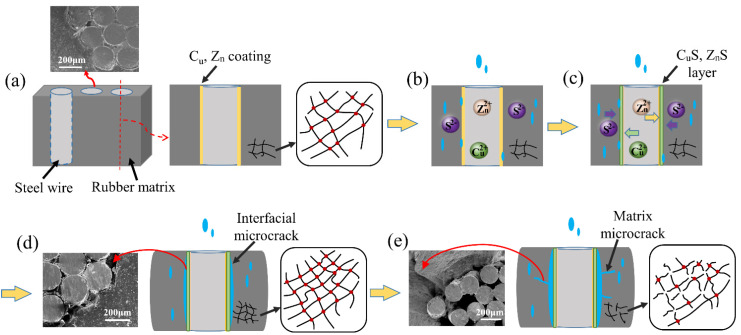
Hydrothermal aging mechanism of ASRT composite. (**a**) Copper plated steel wire and rubber substrate; (**b**) Water absorption; (**c**) Alloy Coating; (**d**) Interface damaged by erosion; (**e**) Matrix microcrack.

**Table 1 polymers-14-03098-t001:** The composition of rubber composite.

Components	Content (%)
Nature rubber	54.69
Carbon black N326	32.54
zinc oxide	4.38
Antioxidant 4020	1.09
Adhesion agent SP1068	1.27
Vulcanizing agent	3.42

**Table 2 polymers-14-03098-t002:** Glass transition temperatures (T_g_) of NR composites under different conditions.

Temperature (°C)	Aging Time (d)			
	0	21	49	84
30	−54.4	−54.3	−53.7	−52.8
60	−54.4	−53.5	−51.8	−53.9
90	−54.4	−54	−54.3	−56.7

## Data Availability

All data are displayed in the manuscript.

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
