# Peer review of "Hydrothermal Aging Mechanisms of All-Steel Radial Tire Composites"

_polymers, 2022, doi:10.3390/polym14153098_

Round 1
Reviewer 1 Report
The Authors respond appropriately to the comments from Reviewers, so I think it is no problem to publish this manuscript in Polymers.
Author Response
The Authors respond appropriately to the comments from Reviewers, so I think it is no problem to publish this manuscript in Polymers.
The authors’ answer: We appreciate the reviewer’s positive evaluation of our work.

Reviewer 2 Report
The manuscript titled “Hydrothermal aging mechanisms of all-steel radial tire composites” discussed the effects of hydrothermal environment on the aging of all-steel radial tire (ASRT) composites. Several properties such as mechanical, thermal, morphological and cross-linking density were studied before and after aging in this work and some positive results were reported in this manuscript. This research has value for the researchers in the related areas. However, the paper needs improvement before acceptance for publication in the Polymers journal. My detailed comments are as follows:
1. How does temperature affect the water absorption properties of the ASRT composites in this work?
2. Authors measured cross-linking density at three different temperatures. At what temperature they got highest cross-linking density and why?
3. Authors should mention the glass transition temperature value of NR and NR composites in the discussion part of 3.3. Section.
4. How does the microstructure of the ASRT composites affect the final properties of these composites?
5. Basically hydrothermal aging leads to reduction in tensile strength, interfacial shear strength of the ASRT composites in this work. So how this hydrothermal aging process can be effective in the practical application of tire composite?
Author Response
The manuscript titled “Hydrothermal aging mechanisms of all-steel radial tire composites” discussed the effects of hydrothermal environment on the aging of all-steel radial tire (ASRT) composites. Several properties such as mechanical, thermal, morphological and cross-linking density were studied before and after aging in this work and some positive results were reported in this manuscript. This research has value for the researchers in the related areas. However, the paper needs improvement before acceptance for publication in the Polymers journal. My detailed comments are as follows:
- The reviewer’s comment:How does temperature affect the water absorption properties of the ASRT composites in this work?
The authors’ answer: The higher aging temperature can lead to the more water absorption of the ASRT composites at same time. This is because the high temperature can promote the molecular movement of water and composites matrix, which further enhanced the uptake of water.
- The reviewer’s comment:Authors measured cross-linking density at three different temperatures. At what temperature they got highest cross-linking density and why?
The authors’ answer: The highest value of cross-linking density is 13.59 E-4mol/ml after aging 49 days at 60°C in the measured date. The cross-linking of the molecular chains are facilitated by temperature and moisture. The high temperature and long aging time can lead to the break of molecular chain which resulting in a decrease of cross-link density.
- The reviewer’s comment:Authors should mention the glass transition temperature value of NR and NR composites in the discussion part of 3.3. Section.
The authors’ answer: The date of glass transition temperature value of NR and NR composites have been added in Table 2. Thanks to you for your good comments.
- The reviewer’s comment: How does the microstructure of the ASRT composites affect the final properties of these composites?
The authors’ answer: The ASRT composites contains three parts of steel wire fiber, matrix and interface. There is no obvious change of microstructure of steel wire fiber in SEM test. While the microcrack appeared in the interface and matrix with the increase of aging time at different temperature. Those microcrack can lead to the decline of mechanical properties and promote the water absorption.
- The reviewer’s comment: Basically hydrothermal aging leads to reduction in tensile strength, interfacial shear strength of the ASRT composites in this work. So how this hydrothermal aging process can be effective in the practical application of tire composite?
The authors’ answer: The aim of our work is to research the effect of hydrothermal aging on ASRT composites mechanical properties and thermo-mechanical performance. The aging behavior and mechanism of ASRT composites in hydrothermal environment were also investigated. All those works will provide mechanism and methods to improve the durability of ASRT composites for application, especially in hydrothermal environment. For example, the surface of steer wire fiber will be treated based our results to improve the interface between fiber and matrix before use.

Reviewer 3 Report
1) Please extend descriptions of the research methods about the sample preparation, size and etc.
2) Figure 4- the authors say that only uma degrades, but is there no corrosion of rebar? The SEM photos show a white coating suggesting the foam formed during corrosion
3)why the authors did not complete the research for 7 days of aging?
4)the results for cross-linking density are quite surprising for the temperature of 30 degrees, the increase is practically linear, the same is for the temperature of 90, but here we have a decrease. What happens at an intermediate temperature of 60 degrees? isn't there some process taking place in the rubber matrix here?
5) Please correct Mpa to MPa on figure 9
6) You measure elongation at break or elongation to break?
7) on Figure 9 the trendline and the quadratic equation are missing for a better understanding. According to the data, there is a certain spread that should be more or less determined by the trend line.
9) of how many trials were the measurement error placed next to the data?
10) There is no comparison of the authors' results with other studies on the aging of rubber composites based on tires and other products. Can the authors add data to the chart that would confirm or contradict the correctness of the data results obtained?
11)The authors also do not mention the calculation of the measurement error. On what basis was it derived, which coefficients were derived, which populations were taken into account, and which results were discarded? Please provide a table or a graph. What statistics of measurement error were made for the tested samples?
Round 2
Reviewer 2 Report
Since the manuscript was improved based on inquiries most of them, this reviewer could suggest it to publish.
Reviewer 3 Report
Thank You for all your responses. At this moment the manuscript is very good and may be processed further. In my opinion, all the problems of the article have been explained and extended. It recognizes that the article is ready for printing in its current form.